# TGF*β* Inhibitor A83-01 Enhances Murine HSPC Expansion for Gene Therapy

**DOI:** 10.3390/cells12151978

**Published:** 2023-07-31

**Authors:** Jenni Fleischauer, Antonella Lucia Bastone, Anton Selich, Philipp John-Neek, Luisa Weisskoeppel, Dirk Schaudien, Axel Schambach, Michael Rothe

**Affiliations:** 1Institute of Experimental Hematology, Hannover Medical School, 30625 Hannover, Germany; fleischauer.jenni@mh-hannover.de (J.F.); bastone.antonella@mh-hannover.de (A.L.B.); selich.anton@mh-hannover.de (A.S.); john-neek.philipp@mh-hannover.de (P.J.-N.); weisskoeppel.luisa@mh-hannover.de (L.W.); schambach.axel@mh-hannover.de (A.S.); 2REBIRTH—Research Center for Translational Regenerative Medicine, Hannover Medical School, 30625 Hannover, Germany; 3Department of Inhalation Toxicology, Fraunhofer Institute for Toxicology and Experimental Medicine ITEM, Nikolai Fuchs Strasse 1, 30625 Hannover, Germany; dirk.schaudien@item.fraunhofer.de; 4Division of Hematology/Oncology, Boston Children’s Hospital, Harvard Medical School, 30625 Hannover, Germany

**Keywords:** hematopoietic stem and progenitor cells, stem cell expansion, A83-01, TGFbeta signaling, mast cells, gene therapy, genotoxicity, IVIM, SAGA

## Abstract

Murine hematopoietic stem and progenitor cells (HSPCs) are commonly used as model systems during gene therapeutic retroviral vector development and preclinical biosafety assessment. Here, we developed cell culture conditions to maintain stemness and prevent differentiation during HSPC culture. We used the small compounds A83-01, pomalidomide, and UM171 (APU). Highly purified LSK SLAM cells expanded in medium containing SCF, IL-3, FLT3-L, and IL-11 but rapidly differentiated to myeloid progenitors and mast cells. The supplementation of APU attenuated the differentiation and preserved the stemness of HSPCs. The TGFβ inhibitor A83-01 was identified as the major effector. It significantly inhibited the mast-cell-associated expression of FcεR1α and the transcription of genes regulating the formation of granules and promoted a 3800-fold expansion of LSK cells. As a functional readout, we used expanded HSPCs in state-of-the-art genotoxicity assays. Like fresh cells, APU-expanded HSPCs transduced with a mutagenic retroviral vector developed a myeloid differentiation block with clonal restriction and dysregulated oncogenic transcriptomic signatures due to vector integration near the high-risk locus *Mecom*. Thus, expanded HSPCs might serve as a novel cell source for retroviral vector testing and genotoxicity studies.

## 1. Introduction

Monogenic disorders, such as primary immune deficiencies [1,2], hemoglobinopathies [3,4,5], bone marrow failure syndromes [6,7,8], and lysosomal storage diseases [9,10], cause life-threatening symptoms. Ex vivo gene correction with retroviral vectors and transplantation of genetically modified cells were effective in clinical trials for such disorders and could reduce disease symptoms. To develop new gene therapy vectors, mouse hematopoietic stem and progenitor cells (HSPCs) serve as crucial model systems for gene modification. Finding expansion protocols for HSPCs would be desirable to reduce the need for animal donors. Ex vivo cultivation outside the bone marrow (BM) niche can lead to rapid differentiation and the loss of long-term engraftable hematopoietic stem cells (HSCs). Thus, improvements in HSC cultivation for gene modification and reliable safety assays of retroviral vectors are essential for the clinical translation of new gene therapy strategies according to 3R principles.

To maintain the stemness of HSPCs after ex vivo gene correction, media with cytokines mimicking the microenvironment of the stem cell niche are the basis of expansion protocols. Specific combinations of key cytokines, such as stem cell factor (SCF), interleukin 3 (IL-3), IL-11, IL-6, and FMS-like tyrosine kinase 3 ligand (FLT3-L), can highly influence the phenotype of the expanded HSPCs [11,12], their proliferation capacity, and their long-term engraftment. Lui and colleagues showed how media supplemented with SCF, IL-3, and IL-6 promoted high numbers of phenotypically short-term HSCs (lineage-negative [lin-] Sca1+cKit+, LSK), with the majority of these cells expressing the Fc epsilon receptor 1 alpha (FcεR1α+), typical for mast cells. By exchanging IL-3 and IL-6 for FLT3-L and IL-11, they significantly reduced the percentage of mast cells in culture. However, this came along with a reduction in LSK cell expansion.

In this study, we initially evaluated four different media and their ability to expand murine BM-derived LSK SLAM (LSK CD48- CD150+) cells. The best candidate contained the cytokines SCF, IL-3, FLT3-L, and IL-11 (S3F11), but expectedly led to the differentiation of myeloid progenitors and mast cells. Thus, we supplemented S3F11 with the small molecules A83-01, pomalidomide, and UM171 (S3F11+APU) to target signaling pathways that induce HSC propagation and counteract quiescence [13]. A83-01 inhibits TGFβ signaling by targeting activin receptor-like kinase 5 (ALK5, TGFβ type I receptor), ALK4 (type I activin/nodal receptor), and ALK7 (type I nodal receptor), which consequently blocks the phosphorylation of Smad2 [14,15]. Since high concentrations of TGFβ promote quiescence in HSCs, we hypothesized that inhibition with A83-01 would lower the levels of TGFβ and allow the cycling of HSCs [16,17]. Additionally, we used pomalidomide to influence Notch signaling, which stimulates HSC self-renewal [18,19]. Notch is negatively regulated by the transcription factor Ikaros [20,21,22]. In the context of multiple myeloma, thalidomide and its derivate pomalidomide could promote the degradation of Ikaros by altering the substrate specificity of cereblon (CRBN), the target of the CUL4-RBX1-DDB1-CRBN (also known as CRL4^CRBN^ E3 ubiquitin ligase. Upon pomalidomide treatment, CRL4^CRBN^ degrades neosubstrates, such as Ikaros, instead of its endogenous targets, such as MEIS2 [23,24]. Furthermore, we used the small molecule UM171, which has proven to maintain stemness and proliferation in human and murine HSPC expansion protocols [25,26,27]. Our data suggest that the TGFβ inhibitor A83-01 primarily maintained murine HSPC attributes and an arrest of differentiation. A83-01 significantly attenuated mast cell differentiation while promoting LSK cell expansion. Thus, we underline the importance of TGFβ in hematopoiesis, especially the effect on the cycling state of HSCs [28].

We tested the functionality of expanded HSPC in two in vitro genotoxicity tests. The in vitro immortalization (IVIM) assay [29] and the surrogate assay for genotoxicity (SAGA) [30] assess the mutagenic potential of retroviral vectors on murine BM-derived lin- cells (Scheme I). The IVIM assay quantifies the outgrowth of insertion mutants after a low cell density replating step. SAGA determines the dysregulation of genes associated with the in vitro transformation of murine HSPCs. Both assay systems rely on primary mouse cells. We were determined to use expanded cell material as input in the assays to reduce the number of animals needed as donors. Despite a reduced frequency of clonal outgrowth, APU-expanded (-Exp) cells harbored characteristic high-risk insertions near the *MDS1 and EVI1 complex locus* (*Mecom*) when transduced with a mutagenic vector, reflecting the main findings of previous IVIMs with nonexpanded cells. It is noteworthy that the immune phenotyping of expanded cells transduced with oncogenic vectors showed a differentiation block of early myeloid progenitors. Safer vector designs did not lead to the immortalization of fresh or expanded cells in IVIM. SAGA revealed a similar enrichment of stemness- and transformation-associated genes after oncogenic vector transduction in APU-Exp and fresh lin- cells.

In summary, our work shows how cells cultivated in A83-01 retain a stem-cell-like phenotype after expansion, making them suitable for testing retroviral vector transduction efficiency, control of the long-term transgene expression, and genotoxicity studies.

## 2. Materials and Methods

### 2.1. LSK SLAM Sorting and Expansion

BM cells were harvested by flushing tibiae, femora, and iliac crests of 8- to 12-week old female C57BL/6J mice (Janvier, Pays de la Loire, France). Erythrocytes were lysed by incubating the BM cell suspension in red blood cell lysis buffer (1.5 M ammonium chloride, 100 mM potassium hydrogen carbonate, and 10 mM disodium EDTA in distilled water with pH = 7.2) for 10 min (1.5 mL/mouse). The reaction was stopped by washing with magnetic-activated cell sorting (MACS) buffer (0.5% FCS (PAN Biotech, Aidenbach, Germany)) and 2 mM EDTA (PAN Biotech) in DPBS (PAN Biotech). Purified cells were depleted for hematopoietic lin markers (CD5, CD45R, CD11b, Anti-Gr-1 (also known as Ly6G), 7-4, and Ter-119) by MACS according to the manufacturer’s protocol (Miltenyi Biotec, Bergisch Gladbach, Germany, Catalogue No. 130-090-858). Before FACS staining, lin-depleted cells were incubated for 15 min with 0.5 μg CD16/CD32 antibody (eBioscience, San Diego, CA, USA, Catalogue No. 14-0161-82) in FACS buffer consisting of PBS (PAN Biotech) supplemented with 2% FCS (PAN Biotech) and 2 mM EDTA (PAN Biotech) to prevent unspecific antibody binding. To obtain lin- cells, lin-depleted BM cells were stained with the “Biotin anti-mouse Lineage Panel” (Anti-CD11b, -GR1 (also known as Ly6g), -Ter119, -CD3e, and -CD45R) provided by BioLegend (Catalogue No. 133307) and coupled to streptavidin BV785 (BioLegend, San Diego, CA, USA, Catalogue No. 405249). LSK SLAM cells were identified by staining with anti-cKit/APC-Cy7 (BioLegend, Catalogue No. 105826), anti-Sca-1/PerCP-Cy5.5 (eBioscience, Catalogue no.45-5981-82), anti-CD48/FITC (BioLegend, Catalogue No. 103404), and anti-CD150/PE-Cy7 (BioLegend, Catalogue No. 115914). Finally, samples were resuspended in 0.2 μg/mL DAPI (Sigma-Aldrich, Saint Louis, MO, USA, Catalogue No. D9542) for dead cell exclusion. FACS was performed with the FACSAria Fusion cell sorter (Becton Dickinson, Franklin Lakes, NJ, USA), and analyses were performed with FACSDiva, Version 8.0.1 software (Becton Dickinson). Sorted LSK SLAM cells or lin-depleted cells were seeded in a 96-well flat-bottom plate for suspension cells (Sarstedt, Nümbrecht, Germany) with a cell density of 50 cells/well in different expansion media: STIF, SFT, S3F11, S3F11T, and S3F11+APU (media and concentrations in Appendix A). Half-medium changes were performed twice weekly, and wells were expanded if confluence reached 70%.

### 2.2. MTT Assay

MTT assay was used to determine the number of expanded cells by correlating the metabolic activity to a standard curve of lin- cells. For the IVIM assay, the absorbance was compared with the microscopical scoring. Thus, cells were stained with 0.25% thiazolyl blue tetrazolium bromide (MTT, Sigma) in DPBS (Pan Biotech) for 2–3 h at 37 °C before lysing the cells in 10% SDS (Sigma). Plates were incubated while shaking overnight at RT. Absorption was measured at 540 nm with the plate reader SpectraMax 340PC (Molecular Devices, San Jose, CA, USA).

### 2.3. Flow Cytometry

The immune surface phenotype was assessed for myeloid, lymphoid, mast cell, and multipotent HSPC marker: Anti-CD3ε/APC-Cy7 (BioLegend Catalogue no.100330), anti-B220-APC/Cy7 (BioLegend Catalogue No. 103224), anti-Ter119/APC-Cy7 (BioLegend Catalogue No. 103224), anti-CD11b/AF700 (BioLegend Catalogue No. 103224), anti-Gr1 (BioLegend Catalogue No. 108422), anti-FcεR1α/FITC (eBioscience Catalogue No. 11-5898-85) or anti-FcεR1α/PE (eBioscience, Catalogue No. 12-5898-83), anti-cKit/APC-Cy7 (BioLegend, Catalogue No. 105826), and anti-Sca-1/PerCP-Cy5.5 (eBioscience, Catalogue No. 45-5981-82). Samples were finally stained within 0.2 μg/mL DAPI (Sigma-Aldrich, Catalogue No. D9542) for dead cell exclusion. Flow cytometry was performed with the Cytoflex S (Beckman Coulter, Brea, CA, USA) cytometer, and data were analyzed with the CytExpert (Beckman Coulter) and FlowJo, Version 10.2 (Tree Star, Ashland, OR, USA) software.

### 2.4. Cytospin Analysis

For microscopic analysis of cytospins, 5 × 10^4^ cells were centrifuged for 10 min at 800 rpm on microscopic slides using the Thermo Shandon Cytospin 4 centrifuge (Thermo Fisher Scientific, Waltham, MA, USA). For the Pappenheim staining, cells were stained 5 min in May-Grünwald dye, washed, and then stained for 20 min in Giemsa dye (Sigma-Aldrich). Cytospins were visualized with a NanoZoomer S210 digital slide scanner (Hamamatsu, Hamamatsu City, Japan) using 40× magnification. Cell morphology was assessed with the NDP.view2 2.8.24 viewing software U12388-01 (Hamamatsu).

### 2.5. In Vitro Immortalization Assay

Cell-culture-based assay was used to determine the genotoxic risk of retroviral vectors as previously described [29,30]. In brief, BM-derived lin- cells were thawed and prestimulated for 2 days in serum-free StemSpan^TM^ SFEM (STEMCELL Technologies, Vancouver, BC, Cananda) supplemented with S3F11 cytokines before transduction (Figure 1).

For this study, cells were modified either with an LTR-driven gammaretroviral vector encoding a fluorophore as a positive control named RSF91 (previously described here [31,32,33]), a SIN lentiviral vector with an EFS promoter (abbreviated as EFS and described here [34,35]), or remained untransduced (mock). On day 15 post-transduction, 100 cells/96-well (flat-bottom suspension plate) were replated for another 2-week culture period. Afterward, plates were screened for wells showing clonal transformation. In addition, the metabolic activity was measured by MTT to determine the number of positive wells showing transformation and the replating frequency.

### 2.6. VCN Determination by qPCR

Vector copy number (VCN) per diploid cell was determined by quantifying the amount of vector-derived woodchuck hepatitis virus post-transcriptional regulatory element (WPRE) compared with genomic polypyrimidine tract binding protein 2 (PTBP2) by Taqman qPCR on an Applied Biosystems StepOnePlus System (Foster City, CA, USA). Thus, 100 ng of genomic DNA from transduced (RSF91, EFS) and mock samples were used per PCR reaction. The ABI Taqman Fast Advanced Master Mix (Thermo Fisher Scientific) was used with 660 nM WPRE and PTBP2 primer and 150 nM of the respective probes. The PCR was run in 40 cycles.

### 2.7. Surrogate Assay for Genotoxicity Assessment

We previously developed the machine learning algorithm “surrogate assay for genotoxicity assessment” (SAGA) to predict the mutagenic risk of viral vectors for clinical studies [30,36]. Cells were harvested for RNA isolation on day 15 of the IVIM assay (Figure 1). Thus, a cell pellet of half of the bulk was frozen in 350 μL RLT lysis buffer (Qiagen). RNA was isolated with the RNeasy kit (Qiagen) according to the manufacturer’s protocol. RNA was transferred to the research core unit genomics (RCUG) of the Hannover Medical School for quality control using the 2100 Bioanalyzer (Agilent) and microarray analysis with the microarray scanner G2565CA (Agilent). Probes were annotated to the Whole Mouse Genome Oligo Microarray 4 × 44 K v2 and mapped to 60mer sequences to a recent release of the murine transcriptome (Gencode version M181, GRCm38.p6, release 07/2018). SAGA analysis was performed with R 4.2.0, Bioconductor 3.15, and the R package “saga” described here: https://talbotsr.com/saga_package/index.html (accessed on 5 July 2023). The whole code can be downloaded from Github (https://github.com/mytalbot/saga_package/, accessed on 5 July 2023). In brief, sample information was given in the “SampleInformation” .txt file before importing microarray .txt files to the SAGA pipeline. Raw data were preprocessed by quantile normalization and batch correction. Transforming and untransforming vectors were discriminated by principal component analysis (PCA) using the support vector machine (SVM), which is based on 20 predictors for genotoxicity. Gene set enrichment analysis (GSEA) revealed a normalized enrichment score (NES), which classifies the enrichment of 11 predictors within transduced samples compared with mock. Safer vectors usually score an NES < 1, whereas mutagenic vectors have a high NES > 1.

### 2.8. Insertion Site Analysis with INSPIIRED

INSPIIRED (integration site pipeline for paired-end reads) is a pipeline developed by Berry and colleagues to determine vector integration sites [37]. Genomic DNA was isolated from IVIM bulk (day 15 or day 21) or clones using the DNeasy Blood and Tissue Kit (Qiagen). DNA was sheared by sonification using the S220 Focused-ultrasonicator (Covaris, Woburn, MA, USA). Purification was performed with AMPure Beads (Beckman and Coulter) in a 0.7-fold ratio. Fragmented DNA was end-repaired with the NEBNext Ultra II End Repair/dA-Tailing Module (New England Biolabs (NEB), Frankfurt, Germany) and ligated with specific linkers using the NEBNext Ultra II Ligation Module (NEB). After following the same purification procedure with the AMPure Beads, nested PCRs were run to amplify the vector-genome junctions by adding Illumina adapter sequences, using specific index primers and sample-specific linker primers. The PCR products were visualized on 2% agarose gels, measured by Qubit, and pooled into DNA libraries that were transferred to the RCUG of the Hannover Medical School for quality control with a Bioanalyzer device and analysis by Illumina sequencing on flow cells with 1 million clusters. Linker and primer sequences used for PCR reactions are available upon request. Downstream bioinformatic processing was generally performed as described by Berry and colleagues 37. The analysis files necessary to run the INSPIIRED pipeline were downloaded from GitHub (https://github.com/BushmanLab/INSPIIRED, accessed on 5 July 2023). Demultiplexed sequences in FASTQ format were generated according to the individual index primers used, quality checked, aligned, and annotated to the mouse genome (mm9). The plasmid vector sequences served as a reference for LTR regions and vector trimming. The processing and alignment statistics were exported before uploading the results to a local database. After creating a sample management database, reports with all integration site data were generated and used for customized postprocessing in Excel, Version 2016 and GraphPad Prism, Version 9 (GraphPad Prism Incorporation, San Diego, CA, USA).

### 2.9. Transcriptome Analysis

RNA was isolated from 2-week expanded material and used for microarray analysis, likewise described above. S3F11-Exp cells were compared with S3F11+A83-01-Exp cells for significantly differentiated cells (*p* < 0.05). The R packages “limma”, “fgsea”, “RColorBrewer”, “gplots”, and “ggplot2” were used to generate heat maps and volcano plots. Gene set enrichment analysis was performed using the ontology M5 gene sets from the Mouse Molecular Signatures Database (MSigDB) and filtered for expected hematopoietic cell types (“hematopoietic”, “mast_cell”, “myelo”, “monocyte, “lymphoid”, “erythro”, “macrophage”, “leuko”, “platelet”, “neutrophil”, “granulocyte”) and key words of the TGFβ signaling pathway (“transforming_growth_factor”, “activin_receptor”, “smad”, “epithelial_to_mesenchymal_transition”, “MAPK”).

### 2.10. Protein Extraction and Western Blot Analysis

Inhibition of TGFβ signaling was analyzed based on the protein level of phospho-Smad2/3. Cells expanded for 2 weeks in S3F11 or S3F111+A/APU were lysed in RIPA buffer (150 nM NaCl, 50 mM Tris-HCl, 1% Nonidet P-40, 0.5% sodium deoxycholate, 0.1% SDS in ddH2O) supplemented with a protease inhibitor (cOmplete^TM^, Mini Protease Inhibitor Cocktail, Roche). Based on the Bradford method, the protein concentration was determined with the Bio-Rad Protein Assay Dye Reagent Concentrate (Catalogue No. 500000630). An amount of 30 μg protein was loaded on a 12.5% SDS polyacrylamide gel and blotted on a nitrocellulose membrane. Phospho-Smad2 was detected by the respective primary antibody (anti-phospho-Smad2, Ser465/467, rabbit polyclonal, 1:500, Sigma-Aldrich, Catalogue No. AB3849-I) and HRP-conjugated anti-rabbit IgG secondary antibody (1:4000, Cayman Chemical Company, Ann Arbor, MI, USA, Catalogue No. 10004301) for detection. Normalization was performed based on the housekeeping protein vinculin (anti-vinculin, mouse monoclonal, 1:20,000, Sigma, V9131) visualized by HRP-conjugated anti-mouse IgG secondary antibody (1:4000, Cayman Chemical Company, Catalogue No. 10004302). The SuperSignal^TM^ West Pico PLUS Chemiluminescent Substrate (Thermo Scientific, Catalogue No. 34580) was added to the membranes for chemiluminescent detection. The Fusion FX7 (Vilber Lourmat) device acquired imaging, and protein bands were quantified based on their densitometry with the FusionCapt Advance Solo 4 16.07 software (Vilber Lourmat, Marne La Vallée, France).

## 3. Results

### 3.1. Massive Ex Vivo Expansion in S3F11+APU Medium after LSK SLAM Purification

We aimed to expand primary murine HSPCs to use them in preclinical gene therapy safety assays. However, increased in vitro proliferation of HSPCs often leads to their differentiation. Hence, we sought to find culture conditions that preserve the HSPC phenotype characterized by the absence of a lineage marker (CD5, CD45R, CD11b, Anti-Gr-1 (also known as Ly6G), 7-4, and Ter-119) and the expression of cKit and Sca-1 (LSK), while increasing the overall cell pool. In our initial screen, we explored four different cytokine combinations (STIF, SFT, S3F11, and S3F11T described in Appendix A) for the ex vivo expansion of BM-derived HSPCs. We compared the expansion potential of seeding either highly purified LSK SLAM cells, likewise described by Wilkinson and colleagues [38], or lin- cells from murine bone marrow in low cell density (50 cells/96-well) (Figure 1A).

HSPC expansion was estimated with a fold change of over 20,000 in S3F11(T)medium based on their metabolic activity in the MTT assay, but only if they were sorted for LSK SLAM (Figure 1B). Sorting only for lin- cells was not sufficient for notable expansion in low cell density. Therefore, HSPCs were always purified for LSK SLAM markers before ex vivo expansion. Since the addition of thrombopoietin (T) resulted in the same expansion as seen in S3F11, it was neglected in the following experiments. Despite the massive expansion, S3F11 medium induced rapid differentiation of LSK SLAM cells towards the myeloid lineage (55.5% CD11b/GR1+ until day 7), followed by downregulation of myeloid markers and accompanied FcεR1α expression (13.8%) until day 14 (Figure 1C). The small compounds APU have been shown to expand human CD34+ cells while maintaining their stemness successfully [13]. We hypothesized that adding APU to the S3F11 medium (S3F11+APU) might decrease the differentiation of murine HSPCs while they continue to proliferate. Indeed, on day 7 of expansion in S3F11+APU, still 88.3% of the population remained lin- in contrast to the significantly reduced lin- population in S3F11 medium (41.5%, *p* < 0.0001, Figure 1C). Even on day 14 of expansion, S3F11+APU-Exp cells still contained 10% more lin- cells than S3F11-Exp cells. Additionally, the presence of myeloid progenitors was significantly reduced by 21.1% in the S3F11+APU medium until day 14 (*p* > 0.01). After 14 days of expansion, 13.8% of S3F11-Exp cells expressed the mast cell marker FcεR1α, which was also significantly attenuated by the addition of APU (<2% FcεR1α+ in S3F11+APU, *p* < 0.0001). The dynamic marker expression is summarized in Figure 1D. Histopathology confirmed the presence of characteristic cytosolic mast cell granules in S3F11-Exp cultures (Figure 1E, black arrows) and the absence in the S3F11+APU medium. The expansion factor was assessed quantitatively using the Casy counting device and showed in addition to the mast cell attenuation an over 200,000-fold expansion of the total cell population, 30,000-fold expansion of lin- cells, around 20,000-fold expansion of lin-cKit+ cells, and 3800-fold expansion of LSK cells in S3F11+APU, significantly higher than in S3F11 only (Figure 1F, *p* < 0.001 for total and *p* < 0.01 for lin-, lin-ckit+, and LSK cells).

### 3.2. A83-01 Is the Key Compound to Attenuate Mast Cell Differentiation

To resolve which compound in the APU cocktail inhibits mast cell differentiation (Figure 1B), we expanded LSK SLAM cells in single and minus-one compound combinations for 2 weeks and monitored the distribution of mast cells, myeloid cells, and lin- cells (Figure 2A–C).

Every condition that harbored A83-01 inhibited the presence of mast cells compared with the basal medium S3F11 (Figure 2A, mean % lin-FcεR1α+ in S3F11-Exp = 22% vs. S3F11+A/AP/AU/APU = 2%, *p* < 0.0001). During mast cell differentiation, S3F11 (and non-A83-01 containing media) cells began to lose the myeloid marker expression in comparison with A83-01-treated cells, which expressed 8–22% more of the myeloid marker CD11b/GR1 (Figure 2B). The abundance of lin- cells was largely unaffected by the supplementation of the compounds (Figure 2C).

### 3.3. Transcriptomic Analysis Confirms the Inhibition of Multiple Differentiation Pathways by A83-01

The transcriptomic signature of S3F11+A83-01-Exp cells was compared with cells expanded in the basal medium (S3F11-Exp). We found 1928 differentially expressed genes after Bonferroni–Holm correction (DEGs; *p* < 0.05, |logFC| > 1), and the technical replicates clustered according to their medium condition (Figure 3A).

Generally, most DEGs were downregulated by A83-01. In Figure 3B, the volcano plot highlights genes on the outer edges with a potentially high biological relevance due to the consideration of statistical significance [–log(adj.pval)] and magnitude of fold change [log(FC)]. Some of the most significantly downregulated genes by A83-01 were mast cell proteases (*Mcpt4*,* Mcpt2*,* Mcpt1*,* Mcpt9* in order of significance) [39,40], chymases (*Cma1*,* Cma2*) [41,42], tryptases (*Tpsg1*,* Tpsb2*,* Tpsab1*) [43,44], and carboxypeptidases (*Cpa6*,* Cpa3*,* Cpa2*) [45,46]—all enriched in mast cell secretory granules [47]. Additionally, we could confirm a significant reduction of FcεR1α (*FceRa*) in A83-01-treated HSPCs on the transcriptional level (differentially expressed genes are listed in Appendix A). We performed a gene set enrichment analysis (GSEA) using the M5 gene ontology gene sets from the Mouse Molecular Signatures Database (MSigDB) to identify potential cell type signatures and dysregulated pathways. First, we filtered for expected hematopoietic cell types (“hematopoietic”, “mast_cell”, “myelo”, “monocyte”, “lymphoid”, “erythro”, “macrophage”, “leuko”, “platelet”, “neutrophil”, “granulocyte”) and TGFβ pathway key words (“transforming_growth_factor”, “activin_receptor”, “smad”, “epithelial_to_mesenchymal_transition”, “MAPK”). The normalized enrichment scores (NES) within the gene ontology sets were plotted against their statistical significance (Figure 3C, all gene ontology pathways in Appendix A and filtered pathways with abbreviations in Appendix A). Most significant gene sets (thrombocytic, myeloid, mast, monocytic, and erythroid cell types) were strongly downregulated in S3F11+A83-01-Exp cells, while only two granulocytic gene sets were significantly enriched. To understand how A83-01 acted on the murine HSPC, we analyzed TGFβ signaling. The ALK5/4/7 inhibitor A83-01 prevents TGFβ signaling by blocking the phosphorylation of Smad2 [14,15]. The phosphorylated Smad2 and Smad3 complex (pSmad2/3) is bound either by TIF1γ to promote erythroid differentiation or by Smad4 to inhibit the proliferation of human HSPCs [48]. On the transcriptional level, we found a significant depletion of gene sets describing TGFβ activation, MAPK cascade, and epithelial-to-mesenchymal transition in A83-01-treated cultures. Western blot analyses confirmed the absence of pSmad2/3 in S3F11+A/APU-Exp cells, but the presence in S3F11-Exp cells (Figure 4A).

From three experiments, A/APU significantly inhibited pSmad2/3 on the protein level (Figure 4B).

### 3.4. Efficient Transduction of S3F11-APU-Exp Cells for IVIM Assay

If expanded cells shall be used in retroviral vector testing, it is pivotal that they still proliferate after efficient gene modification. In the IVIM assay, transduced and nontransduced mock cultures are cultured for 2 weeks in S3F11 medium (Scheme I). Hence, we transduced cryopreserved expanded cells (S3F11-Exp and S3F11+APU-Exp) and fresh lin- cells (standard material) with a mutagenic gammaretroviral vector (RSF91, Figure 5A) or a safer self-inactivating lentiviral vector with the internal elongation factor 1 alpha short promoter driving transgene expression (EFS, Figure 5A).

Before transduction, the immune phenotype of the fresh and expanded material was assessed and compared. In each of the three IVIM assays, the S3F11+APU-Exp cells looked immune phenotypically similar to the fresh lin- cells despite the 2-week-long expansion (Appendix A). S3F11-Exp cells were further differentiated than fresh lin- cells. Fresh lin- cells proliferated immediately after thawing and transduction (Figure 5B, green line). In contrast, S3F11-Exp (orange line) and S3F11+APU-Exp (blue line) cells needed time to recover from thawing. However, before replating, S3F11+APU-Exp cells outcompeted fresh cells regarding their proliferation, while S3F11-Exp grew the least and with lower viability in some experiments (Appendix A). Transduction efficiency was determined by the level of transgene expression (Figure 5C, upper panel) and the vector integration load (Figure 5C, lower panel). For IVIM, we aim for three RSF91-vector integrations per genome and at least 80% transduction efficiency to elicit immortalization reliably. Flow cytometry revealed highly efficient transduction of fresh lin- and S3F11+APU-Exp cells with RSF91 (>95% transgene positive). On the contrary, S3F11-Exp cells reached significantly lower transduction levels with RSF91, probably due to the lack of cell proliferation, crucial for gammaretroviral vectors for a successful nuclear entry [49,50]. The safer design SIN.LV.EFS exhibited overall lower transduction efficiencies than the gammaretroviral vector. However, our results showed a tendency for both expanded cultures towards higher transduction levels compared with fresh lin- cells, with S3F11-Exp displaying the strongest effect (*p* < 0.05). Fresh lin- and S3F11+APU-Exp cells harbored more than three RSF91-vector copies in all samples per diploid cell, whereas more than half of the S3F11-Exp samples’ vector copy number (VCN) was below three. In line with the transgene expression in SIN.LV.EFS-transduced samples, the VCN values in S3F11-Exp cells were significantly higher (*p* < 0.05).

### 3.5. Mutagenic Vector-Induced Differentiation Arrest of Lin- and Myeloid Progenitors in Fresh and S3F11+APU-Exp Cells

To further evaluate expanded cell material for use in genotoxicity assays, we assessed myeloid (CD11b/GR1+), lymphoid/erythroid (CD3/B220/Ter119+), and mast cell (FcεR1α+) differentiation markers by flow cytometry on day 4 (Figure 6A) and day 15 (Figure 6B).

During long-term culture in S3F11, we observed the effect of nonspecific analytical antibody binding by mast cells. Here, we want to highlight the importance of accurate controls to determine specific signals in expansion cultures. Initially, we gated using the fluorescence minus one (FMO) controls. However, when we added the respective isotype control instead of only leaving the specific antibody out, a broad dim false-positive population appeared, especially for anti-CD3/B220/Ter119_APC-Cy7 (Appendix A). We adjusted the gating retrospectively according to the isotype controls (Appendix A). Early cultures (LSK SLAM expansion, day 7) did not harbor any mast cells and thereby no nonspecific binding yet (Appendix A). We could even inhibit this phenomenon in long-term cultures (IVIM, day 29) by incubating the cells with heparin as such false-positive antibody stainings were already described for mast cells on account of their granules (Appendix A) [51,52]. Although fresh lin- cell isolations were cryopreserved with a purity of 70–80%, already on day 4, most cells (>70%) expressed myeloid markers CD11b/GR1+ (Figure 6A, left panel). S3F11+APU-Exp cells had a very similar immune phenotype (>80% CD11b/GR1+) compared with fresh cells (Figure 6A, right panel). In contrast, S3F11-Exp cells expressed only <30% myeloid markers with an increased contribution of mast cells (Figure 6A, middle panel, 40–50% lin-FcεR1α+ cells). After 15 days of cultivation in the S3F11 medium, fresh and expanded cultures mostly contained lin-FcεR1α+ mast cells (Figure 6B). After 3 weeks of cultivation, virtually all nonimmortalized cultures had only FcεR1α+ cells (Figure 6C). Sample RSF91_APU#18_ was immortalized (as later shown by the replating step and integration site analysis) and contained a dominant (60%) lin-FcεR1α- population (Figure 6C, right panel). This immune phenotype was accompanied by high proliferation (Appendix A). On day 15 post-transduction, cells were replated under limiting dilution to allow the proliferation of potential insertional mutants. However, compared with previously performed IVIM assays (meta-analyses, fresh), the mean replating frequency (RF) of fresh RSF91-transduced samples from the current study was lower than expected. Still, 5 out of 12 replicates scored positive (Appendix A, circular dots). When we replated after 22 days of culture instead, the number of positive wells increased to 7 out of 12 (triangle dots). The S3F11-Exp sample RSF91_S3F11#13_ (IVIM20210311) showed one immortalized clone after replating on day 21. For APU-Exp cells, only RSF91_APU#18_ (IVIM20220311) showed the replating phenotype after day 15 and day 21, but with a higher frequency of positive wells. When expanding clones from the 96-well plate, they were at least 50% negative for lin- markers, <5% FcεR1α+, and expressed only 2–43% myeloid markers. The immortalized clonal phenotype was similar among fresh APU-Exp cells and S3F11-Exp cells (Figure 6D).

### 3.6. *Mecom* Integrations in Expanded Material Reproduce Clinical Observations and Previous IVIM Results

Immortalized samples (clones) and the respective bulk cultures (day 15) were further characterized for vector integration sites using the integration site pipeline for paired-end reads (INSPIIRED) [37,53]. Integrations in or near *MECOM* were frequently found in patients with severe adverse events (SAE) after gene therapy with retroviral vectors [54,55,56,57]. Similarly, vector integrations in or near *Mecom* are typically found in RSF91-immortalized samples from IVIM/SAGA assays. For the S3F11-Exp and the S3F11+APU-Exp samples with clonal outgrowth on the 96-well plate, INSPIIRED detected *Mecom* as part of the top 10 most abundant insertions in the expanded clones (7.34% contribution for S3F11-Exp, Figure 7A; 20% contribution for S3F11-APU-Exp, Figure 7B).

This dominant insertion, together with the final clonal composition after replating, was not evident at the time of seeding (day 15) for either cell source. Besides *Mecom*, other integrations near or within leukemia-related proto-oncogenes or tumor suppressors were found within the top 10 most abundant in the expanded clones (*Anxa659* [58], *Nemf* or *Sdccag1* [59,60], *Ldhc* [61,62], *Nbea [63], Rap1gap2* [64,65], *Iqgab1* [66,67], *Cript* [68], *Kalrn* [69], *Pkig* [70,71], and *Ubr*2 [72,73]). For the immortalized culture RSF91_APU#18_, INSPIIRED detected five integrations in or near *Mecom, Cript, Kalrn, Pkig, and Ubr2* with an equal contribution in five clones expanded from the 96-well plate used for replating. This clone with five integrations was also observed when the bulk culture was analyzed after 21 days of cultivation (Figure 7B). The count of unique integration sites (UIS) was highly polyclonal in the bulk samples of day 15 (>4000 UIS) and drastically reduced in the clones (<100 UIS). However, our results suggest that clonal dominance can also be observed in bulk cultures of expanded material after a longer cultivation time.

### 3.7. SAGA Detects an Oncogenic Gene Signature in Expanded Cells after Gammaretroviral Transduction

We previously showed that mutagenic retroviral vectors promote a unique oncogenic gene expression signature compared to mock or safer vector designs that can be used as a more sensitive and reliable alternative than IVIM to predict the genotoxic risk of clinical vectors [30]. SAGA uses a support vector machine (SVM) to classify the samples as transforming based on a core set of eleven predictors. The SAGA bioinformatics pipeline also evaluates the oncogenic signature’s enrichment in transduced samples compared with mock with the gene set enrichment analysis (GSEA) tool. From meta-data analyses, we defined a normalized enrichment score (NES) > 1 as a threshold for mutagenicity. The SAGA-GSEA of all three cell sources revealed a mean NES > 1 for RSF91-transduced samples (Figure 7C, red). For the safer SIN.LV.EFS vector, we would expect an NES < 1 as we observed it in our latest meta-data analysis [30]. Indeed, fresh cells resulted in significantly lower NES for the safer vector than RSF91 (*p* < 0.001). Using expanded cells for SIN.LV.EFS transduction slightly enriched the SAGA core set genes (mean NES_fresh_ = −0.48 vs. NES_S3F11-Exp_ = 0.98; NES_S3F11+APU-Exp_ = 0.23). Although there was no significant difference between NES values for SIN.LV.EFS and RSF91 in both expanded conditions, Cohen’s d revealed a large effect size between mutagenic and safer vector designs using S3F11+APU-Exp cells, which even outperformed the fresh cells (Figure 7D). When S3F11+A-Exp were also tested in IVIM and SAGA, they were efficiently transduced (Appendix A) and gave similar SAGA results, likewise S3F11+APU-Exp cells (Appendix A), but no myeloid differentiation block (Appendix A).

For the IVIM assay, we usually require 5 × 10^5^–1 × 10^6^ lin- cells per experiment. Thus, we isolate lin- cells from the BM of a pool of mice and cryopreserve these in 5 × 10^5^ lin- cells vials. Each mouse harbors around 1 × 10^6^ to 2 × 10^6^ lin- cells, which equal 2–4 vials (Figure 7E). In past experiments, we could isolate between 586 and 1812 LSK SLAM cells per mouse (mean of n = 4 isolations was 1115 cells). If all LSK SLAM cells from one mouse were seeded for expansion in S3F11+APU, we could hypothetically expand them by 201,541-fold (mean expansion factor of total cells shown in Figure 1F) and produce around 449 vials of 5 × 10^5^ HSPCs as a homogenous cell product in one huge batch.

Taken together, RSF91-transduced S3F11+APU-Exp cells evidenced arrested cell differentiation of early progenitors, harbored typical high-risk insertions near *Mecom*, and dysregulated specific genes connected to oncogenesis. The similar behavior of this S3F11+APU-Exp material to fresh cells makes it potentially suitable for genotoxicity assays and thereby reduces the number of experimental mice needed, according to the 3R principles.

## 4. Discussion

Ex vivo HSPC cultivation is currently unavoidable for the retroviral gene correction of HSPCs. In gene therapeutic protocols, IL-3 is part of various transduction and expansion media to promote proliferation [74,75,76,77]. In our study, media containing IL-3 and other cytokines essential for stem cell expansion (S3F11) massively expanded myeloid progenitors, which, however, unfortunately, rapidly differentiated into mast cells. We aimed to attenuate such myeloid differentiation to enable the use of expanded cells for genotoxicity studies. By intervening in the phosphorylation of Smad2 with the small compound A83-01 [15], myeloid differentiation, specifically to mast cells, was inhibited while we could expand lin-cKit+ hematopoietic progenitors over 20,000-fold and LSK cells around 3800-fold.

TGFβ signaling determines self-renewal or differentiation in HSCs. High concentrations of TGFβ can be found in the bone marrow niche and restrict the cycling of HSCs [78,79,80], while low concentrations promote the expansion of myeloid progenitors [28,81]. In other studies, the activation of “quiescent” cells was observed when neutralizing antibodies against TGFβ were combined with myeloid cytokines, such as IL-3 and IL-6. The combinatorial approach increased the number of colony-forming units and the long-term repopulating capacity of HSPCs compared with cells exposed to TGFβ [82,83]. We could observe similar beneficial effects of using the TGFβ inhibitor A83-01 in our S3F11 basal medium for HSPC expansion. However, although the single and minus-one experiments and the microarray analysis suggested that A83-01 is the main effector, we cannot entirely exclude synergistic effects with the other compounds (pomalidomide and UM171).

The HSPC expansion effect of TGFβ inhibition could be of interest for the gene therapy of Fanconi anemia (FA) patients. Due to the BM failure in FA, isolating sufficient cell numbers for gene correction can be challenging [84]. In an ongoing clinical trial (ClinicalTrials.gov, NCT03157804; European Clinical Trials Database, 2011-006100-12), the efficacy of gene correction was proven, but only patients transplanted with higher numbers of transduced CD34+ cells showed substantial hematopoietic reconstitution [85]. The ex vivo expansion of HSPCs might lead to more polyclonal repopulation after gene therapy. Furthermore, TGFβ signaling is described to be overexpressed in FA patients. This effect might contribute to the dysfunctionality of HSPCs in FA patient BM [86]. TGFβ inhibitors were tested as an alternative therapy to combat HSC exhaustion [87] but not combined with a gene therapeutic approach so far.

S3F11+APU-Exp cells could be transduced as efficiently as fresh BM lin- cells. The similar immune phenotype of fresh and S3F11+APU-Exp cells on the day of transduction suggests that early progenitors and potentially true transplantable HSCs can be transduced. Although we did not perform transplantation studies with the expanded mouse material, we are currently adapting the protocol for use with human umbilical cord blood-derived HSPCs to analyze the effects of expansion on engraftment. Xenotransplantation studies with a clinically relevant cell type will shed further light on the applicability of TGFβ inhibition in transduction protocols and, ultimately, hematopoietic stem cell transplantation.

Preclinical safety studies, including genotoxicity testing, of novel integrating vectors for gene therapy are mandatory [88,89]. We have previously developed the genotoxicity assays IVIM and SAGA to assess the risk of insertional mutagenesis by retroviral vectors [29,30]. Both assays constantly require murine BM-derived lin- cells. In this study, we propose using expanded cells to reduce the number of experimental animals while maintaining the quality of the assays. The IVIM assay focuses on the proliferation of transformed cells at very low cell densities. The immune phenotype of immortalized clones was not characterized in detail until now. Here, we presented a promising positive genotoxic readout in the form of a myeloid differentiation block in fresh lin- and S3F11+APU-Exp cells transduced with the mutagenic RSF91 vector. The absence of FcεR1α and a lin- marker but an expression of early myeloid marker GR1/CD11b predicted transformation prior to replating in limiting dilution. This shortened the assay duration by 1 to 2 weeks. Mock and SIN.LV.EFS-transduced cells did not show the differentiation block. However, we observed a decreased incidence of immortalized clones compared with the meta-analysis of the past decades. We could only observe one clear transformation in three assays using S3F11+APU-Exp, and so far, none in the two assays using S3F11+A-Exp cells. Hence, the sensitivity of IVIM with expanded cells might be reduced compared with fresh cell material. So far, we cannot say which progenitor type is required for immortalization. It is likely that fresh lin- cells and expanded LSK SLAM cells contain different progenitors. APU promoted the expansion of multipotent progenitors (MPPs). When the cells are used in the IVIM assay, the frequency of MPPs could influence the rate of vector-induced immortalization. Since insertional mutagenesis requires proliferative pressure to allow the development of clonal dominance, postponing the replating for 1 week increased the number of positively scored wells and even only allowed the development of one clone in IVIM20210311 with S3F11-Exp after longer culture before replating. In long-term monitoring trials, adverse events can occur years after gene correction, which is unfeasible to recapitulate in a cell culture system. Therefore, the detection of vector integration sites and the use of more sensitive assays based on transcriptional dysregulation add necessary measures for reliable prediction of genotoxicity.

*Mecom* is transcribed in HSCs and encodes for transcription factors regulating their proliferation, long-term repopulating capacity, and self-renewal [90,91,92,93]. All transformed clones in our experiments with expanded material harbored an integration in or near *Mecom*. Typically, vector integrations near *Mecom* are responsible for insertional mutagenesis in IVIM/SAGA due to the overexpression of the transcription factors promoting proliferation, as seen in leukemia (reviewed here [94]). Clones derived from RSF91_APU#18_ culture had five codominant insertions per cell. We did not observe any cloncal dominance on day 15, and on day 21, the abundance of each dominant integration was <10%. When patients in gene therapy trials are monitored, the abundance of individual integrations above a certain threshold (previously 30% or now often 10%) can elicit closer monitoring intervals. Where a single integration did not contribute more than 10% in our in vitro example, the combination of the five integrations contributed over 25%. Such examples highlight the immense awareness required for monitoring high-risk insertion sites and the diversity of clonal patterns contributing to clinical trial decision processes.

SAGA unfailingly classified RSF91-transduced samples as mutagenic due to the enrichment in stemness- and cancer-associated genes, regardless of the cell source used in this study. The large effect size of NES_RSF91_ compared with NES_SIN.LV.EFS_ confirmed the clear distinction between mutagenic and the safer vector in fresh and even higher in S3F11+APU-Exp cells. Although the machine learning process for SAGA was developed with fresh cells, we could apply the readout to S3F11+APU-Exp cells. The NES_SIN.LV.EFS_ of S3F11+APU-Exp cells even overlapped less with NES_RSF91_, which resulted in a higher effect size. However, the lowest NES_SIN.LV.EFS_ of the fresh samples was classified around −2 compared with the expanded samples, which only reached a minimum NES of around −1, resulting in a lower mean for the fresh samples. Potentially, the background of cells transduced with a safer lentiviral vector could be higher in S3F11+APU-Exp cells since we did not see signs of transformation on the immune phenotypical level in these samples. S3F11-Exp cells gave the least clear result, likely due to the further progressed differentiation before transduction. On day 15 of IVIM, this could have resulted in a transcriptomic signature too different from the SAGA signature set using fresh cells. The development of the SAGA pipeline used over 160 data points to determine a reliable set of genotoxicity predictors. A similar number of data points would have to be gathered with expanded cell material over time to reach a similar performance. Nevertheless, SAGA’s ability to predict transformation confirmingly showed a higher sensitivity than the IVIM assay.

So far, reliable human hematopoietic genotoxicity assays are missing. The transduction and long-term maintenance of human CD34+ HPSCs is challenging. For our positive control, the gammaretroviral vector RSF91, cell division is crucial for vector integration [49,95,96]. Hence, low cycling of HSPCs limits the high transduction efficiency needed for a positive genotoxic readout. We could not achieve desired transduction levels in CD34+ cells using well-known small compounds for HSPC expansion, StemRegenin-1 or UM171 [25,27]. Thus, we are currently testing the APU compounds to expand human umbilical cord blood-derived HSPCs for a humanized version of the IVIM assay. Characterizing a transforming immune phenotype and dysregulated transcriptional pattern after retroviral transduction could establish a genotoxic readout on transduced patient material before transplantation.

## 5. Conclusions

This study proposes using A83-01 to inhibit mast cell differentiation for mouse HSPC cultured in the presence of IL-3. A83-01 blocked the phosphorylation of Smad2 and led to a 20,000-fold expansion of lin-ckit+ cells and a 3800-fold expansion of LSK cells. Expanded progenitors were transduced efficiently with gamma- and lentiviral vectors comparable to uncultivated lin- cells. We tested their functionality in the preclinical biosafety assays IVIM and SAGA. Expanded HSPCs could immortalize after transduction with a mutagenic vector due to virus integration nearby Mecom, however, with an overall lower incidence than fresh lin- cells. More replicates are required to validate the use of expanded HPSCs in genotoxicity assays, which would reduce the number of experimental animals by 200–400-fold due to the immense expansion potential LSK SLAM cells have in S3F11+A83-01 or +APU medium.

## Figures and Tables

**Scheme 1 cells-12-01978-sch001:**
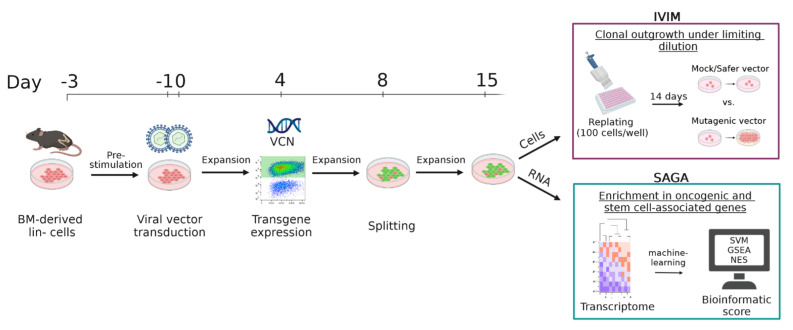
The In Vitro Immortalization (IVIM) and Surrogate Assay for Genotoxicity Assessment (SAGA). Graphical illustration of the standard operating protocol for the IVIM assay and SAGA. Murine BM-derived lin-depleted cells were thawed, prestimulated, and transduced in two rounds with retroviral vectors. Transgene expression and vector copy number (VCN) were determined for transduction efficiency on day 4 post-transduction. After several days of expansion and one splitting step, cells were replated on day 15 in low cell density (100 cells/well) to detect immortalization events in vitro by determining clonal outgrowth after 14 days. RNA was harvested for transcriptome analysis via microarray. Machine learning predicted a classification of vector mutagenicity by a support vector machine (SVM), gene set enrichment analysis (GSEA), and a normalized enrichment score (NES).

**Figure 1 cells-12-01978-f001:**
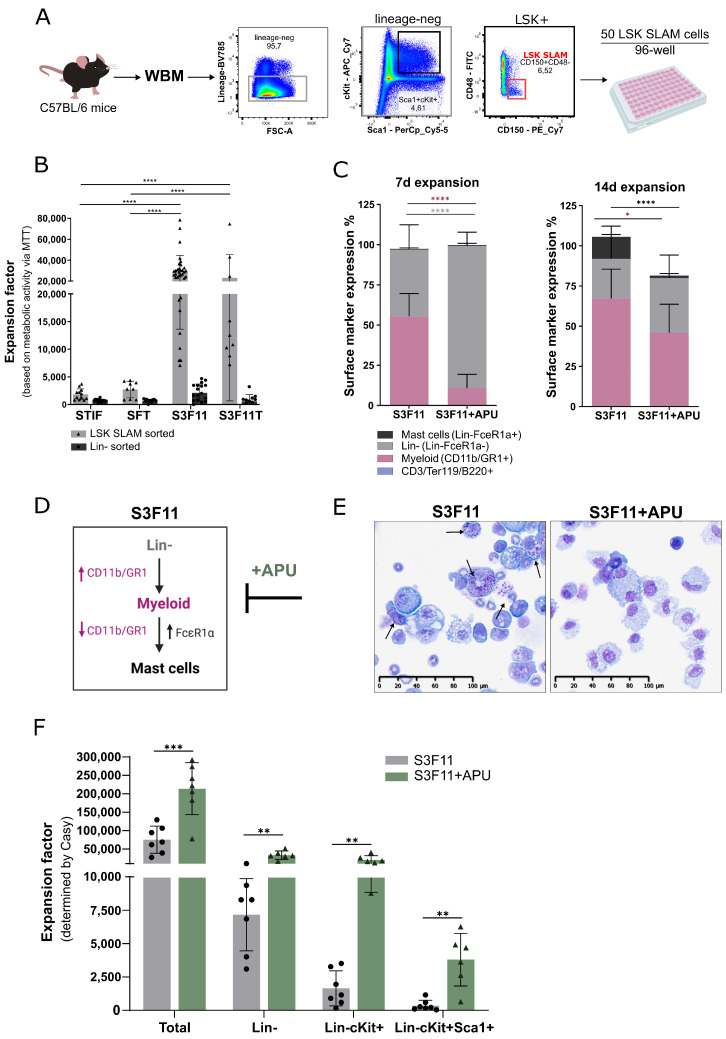
Massive ex vivo expansion after seeding LSK SLAM cells for 14 days in S3F11+APU medium. (**A**) Murine whole bone marrow (WBM) was depleted for cells expressing hematological lin markers (CD5, CD45R, CD11b, Anti-Gr-1, 7-4, and Ter-119) and enriched for LSK (lin-Sca+cKit+) and SLAM (CD48- CD150+) markers. LSK SLAM cells were cultured in a low cell density setting (50 cells/96 well). (**B**) Expansion factor of HSPCs after cultivating 50 LSK SLAM or 50 lin- cells in different cytokine conditions. Cell number was estimated based on metabolic activity measured with MTT dye. A standard curve with a set number of cells was performed for extrapolation. After 2 weeks of expansion in STIF, SFT, S3F11, or S3F11T medium, only medium containing S3F11(T) cytokines led to significantly higher expansion (over 20,000-fold, n = 3) if HSPCs were purified for an LSK SLAM marker. Statistical analysis performed with two-way ANOVA and Bonferroni multiple comparisons comparing media within sorted population. STIF: SCF, TPO, IGF2, FGF2 in StemSpan SFEMI; SFT: SCF, FLT3-L, TPO in StemSpan SFEMI; S3F11: SCF, IL-3, FLT3-L, IL-11 in IMDM+10% FCS; S3F11T: SCF, IL-3, FLT3-L, IL-11, TPO in IMDM+10% FCS. (**C**) Mean (n = 3) of surface marker contribution after 7 and 14 days in the respective medium. Adding APU (A83-01, pomalidomide, UM171) to the S3F11 medium maintained the presence of lin- cells, restrained myeloid (CD11b/GR1+) cells on day 7, and attenuated FcεR1α expression until day 14. Statistical analysis performed with Mann–Whitney test. (**D**) Graphical illustration of the dynamic marker expression during mast cell development in S3F11 medium and the inhibitory influence of A83-01. (**E**) May-Grünwald Giemsa staining of expanded cells on day 14. The arrows indicate representative cells with dark cytosolic granules typical for mast cells. 20× magnification, 100 μm scale bar. ((**F**) Expansion factor (number of expanded cells divided by seeded number of 50 LSK SLAM cells) of total, lin-, lin-cKit+, and lin-cKit+Sca1+ cells after 2 weeks in S3F11 or S3F11+APU medium (n = 3). Cell numbers were assessed by Casy. Significantly more total, lin-, lin-cKit+, and LSK+ cells were expanded in S3F11+APU. Two-way ANOVA with Tukey’s correction for multiple testing. “n” is the number of biological replicates; error bars correspond to SD; *p* < 0.05 = *, *p* < 0.01 = **, *p* < 0.01 = ***, *p* < 0.0001 = ****, and without label if *p* > 0.05 = not significant.

**Figure 2 cells-12-01978-f002:**
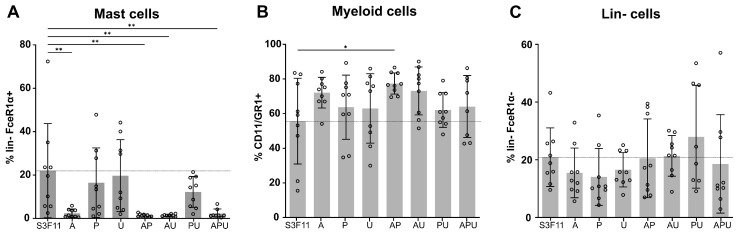
A83-01 is the key compound to attenuate mast cell differentiation in S3F11 medium. Surface marker expression after LSK SLAM expansion in S3F11 (ctrl), S3F11+APU, and each single or minus-one compound combination: (**A**) mast cell (lin-FcεR1α+) (**B**) myeloid (CD11b/GR1+) and (**C**) lin- (lin-FcεR1α-) marker. Dotted line is equivalent to the mean of the control (S3F11). Significantly fewer mast cells were measured in S3F11+A-containing medium. One-way ANOVA with Dunnett’s multiple comparison was applied to n = 3 experiments each in technical triplicate (*p* > 0.05 = *, *p* < 0.01 = **). All compound conditions were compared with the control medium (S3F11).

**Figure 3 cells-12-01978-f003:**
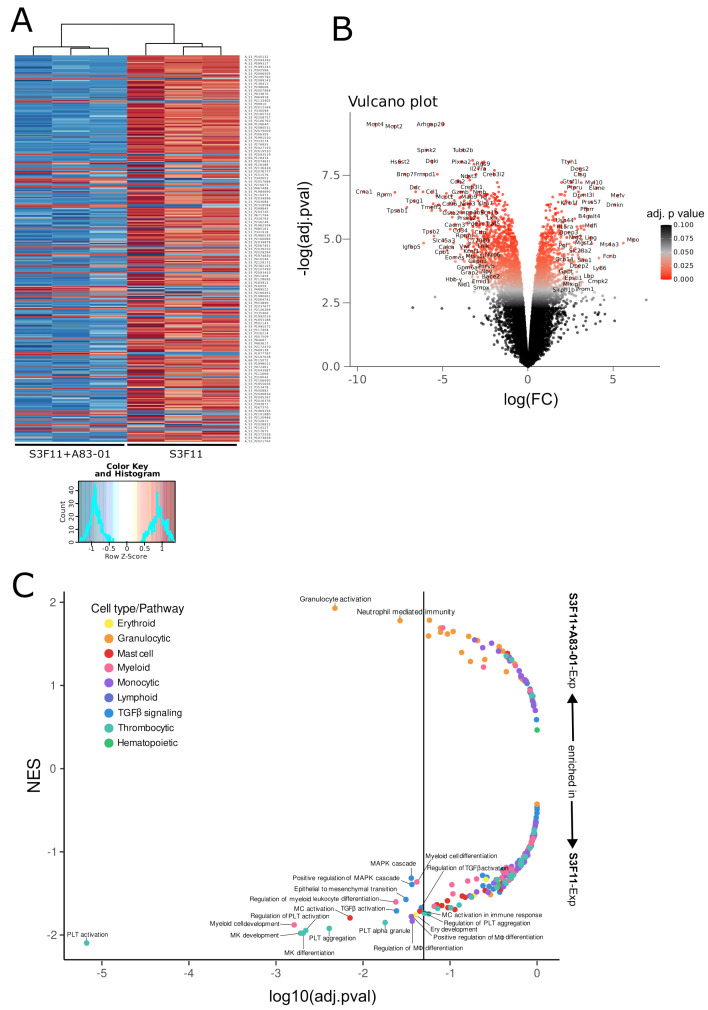
Transcriptomic downregulation of several differentiation pathways by A83-01. Transcriptome analysis of S3F11+A83-01-Exp cells versus S3F11-Exp cells with the Agilent Gene Expression Microarray Platform. (**A**) Heat map with hierarchical clustering of all significantly differentially expressed genes (*p* < 0.05) and strong downregulated gene expression in S3F11+A83-01-Exp cells visualized by the negative row Z-score in blue. (**B**) Volcano plot displaying the negative logarithm of the adjusted *p*-value [-log(adj.pval.)] against the logarithm of the fold change [log(FC)] in S3F11+A83-01-Exp cells compared with S3F11-Exp. (**C**) Gene set enrichment analysis using the ontology gene sets from the Mouse Molecular Signatures Database (MSigDB) and filtered for cell types and pathways of interest. Normalized enrichment score (NES) is plotted against the decimal logarithm of the adjusted *p*-value [log10(adj.pval)]. Significantly enriched/de-enriched gene sets were labeled in abbreviated form. Positive NES equals an enrichment in S3F11+A83-01-Exp cells, while a negative NES equals an enrichment in S3F11-Exp cells. Thrombocytic-, myeloid-, mast-cell-, monocytic-, erythroid-, and TGFβ-signaling-associated gene sets were significantly downregulated by A83-01.

**Figure 4 cells-12-01978-f004:**
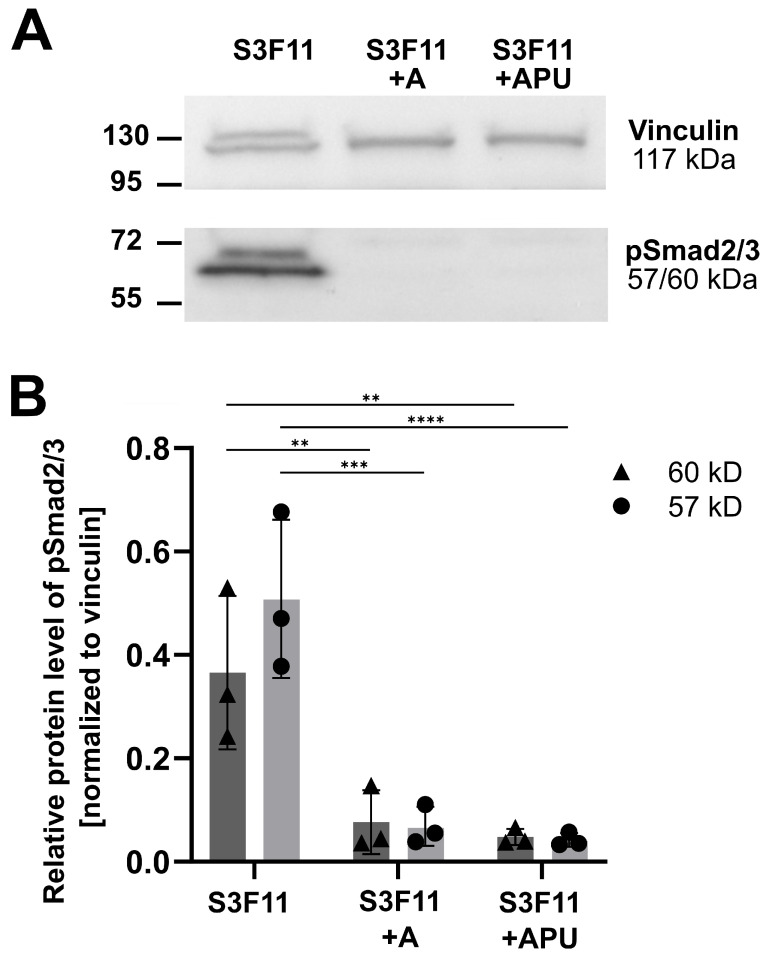
A83-01 inhibits pSmad2/3 expression on the protein level. (**A**) Exemplary Western blot showing the loading control vinculin (117 kDa) and pSmad2/3 (57/60 kDa). Expression of pSmad2/3 is absent in S3F11+A/APU but present in the control S3F11. (**B**) Densitometric quantification of n = 3 blots. Relative protein level of pSmad2 was calculated by normalizing the density of both pSmad2/3 bands to vinculin. Significant reduction of the pSmad2/3 protein level in S3F11+A/APU. Two-way ANOVA with Dunnett’s multiple comparison test was applied to compare A/APU-treated with S3F11 (control), *p* < 0.01 = **, *p* < 0.01 = ***, *p* < 0.0001 = ****.

**Figure 5 cells-12-01978-f005:**
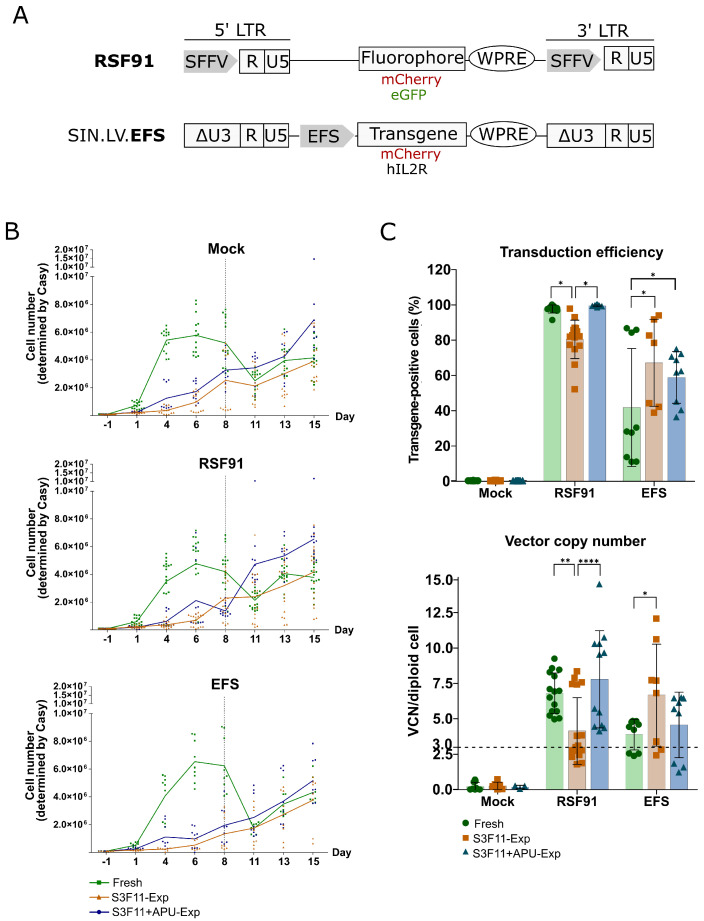
Proliferation capacity and transduction efficiency of fresh lin- cells, S3F11-, and S3F11+APU-Exp cells after retroviral transduction for the IVIM assay. (**A**) Cells were modified either with RSF91, an LTR-driven gammaretroviral vector encoding a fluorophore and used as a positive control, or with a self-inactivating lentiviral vector with a transgene (mCherry or hIL2RG) driven by the elongation factor 1 alpha short promoter (SIN.LV.EFS) connected to the anti-silencing CBX3 element1. (**B**) The proliferation of fresh (green), S3F11-Exp (orange), and S3F11+APU-Exp (blue) cells was compared depending on the vector used for transduction: mock, RSF91-, or EFS-transduced. If possible, cells were split on day 8 by seeding 1 × 10^6^ cells. Cell numbers were determined by Casy and plotted as individual replicates. The line represents the mean. Fresh cells proliferated strongly after transduction, whereas the expanded cells only recovered after day 8 from the thawing and transduction process. S3F11+APU-Exp cells outperformed both fresh and S3F1-Exp cells by day 15. (**C**) Transduction levels (upper panel) and VCN (lower panel) were determined on day 15 as it was the first time point with a sufficient amount of cells. Transgene-positive cells were determined by flow cytometry. VCN per diploid cell was determined by quantifying the amount of vector-derived WPRE in comparison with genomic PTB2 by Taqman qPCR. The dotted line indicates the minimum VCN of three for the reliable transformation of our positive control. Fresh and S3F11+APU-Exp cells were efficiently transduced with RSF91 in contrast to S3F11-Exp. Statistical analysis performed with two-way ANOVA and Tukey´s multiple comparisons test on n = 3 IVIM assays, *p* < 0.05 = *, *p* < 0.01 = **, *p* < 0.0001 = ****.

**Figure 6 cells-12-01978-f006:**
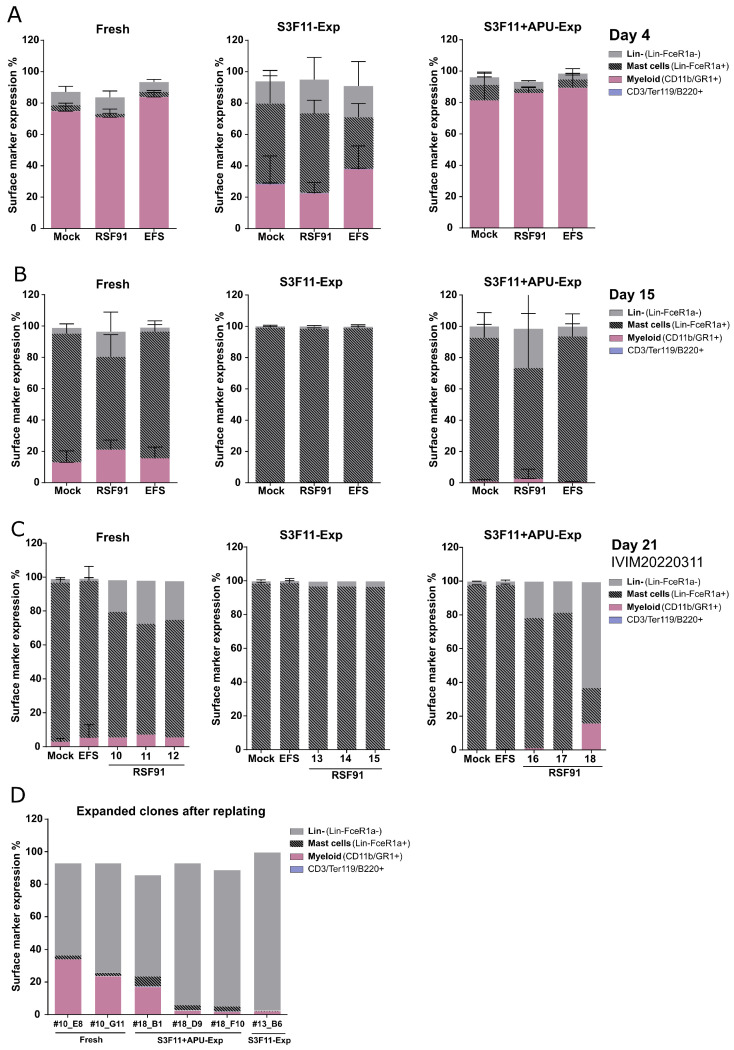
Immune phenotypic characterization reveals differentiation arrest of early myeloid progenitors after gammaretroviral transduction. Samples from all three sources were analyzed at different time points by flow cytometry for their surface marker expression. (**A**) Immune phenotype on day 4 revealed a strong similarity of Fresh and S3F11+APU-Exp cells regarding their composition of lin-, myeloid (CD11b/GR1+), CD3/B220/Ter119+, and mast cell (FcεR1α+) progenitors. S3F11-Exp cells harbored an increased amount of lin-FcεR1α+ cells (n = 3 IVIMs). (**B**) On day 15, in all three cell sources (Fresh, S3F11-Exp, S2F11+APU-Exp), the bulk culture is mostly lin-FcεR1α+ (n = 3 IVIMs). (**C**) Single sample analyses from IVIM 20220311 showing the development of individual RSF91-transduced samples in comparison with mock and EFS after 21 days in culture. RSF91-transduced Fresh lin- and S3F11+APU-Exp cells differ from mock and EFS. Sample ID is mentioned below the bars. (**D**) From positively scored plates, individual wells (labeled below) were picked for the expansion of potential clones in culture. Most clones remained lin-FcεR1α- or CD11b/GR1+.

**Figure 7 cells-12-01978-f007:**
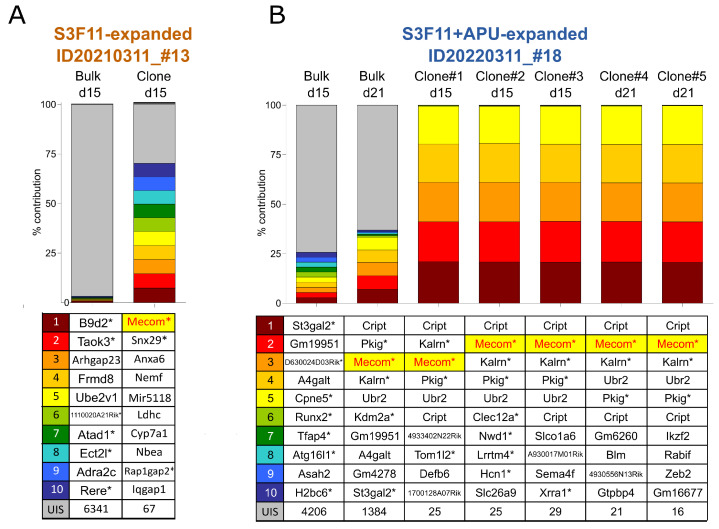
*Mecom* integration and dysregulated oncogenic gene expression in S3F11 and S3F11+APU-Exp cells after gammaretroviral transduction with RSF91. Integration site analysis with the INSPIIRED pipeline and representation of the top 10 most abundant integration sites in (**A**,**B**) with highlights on *Mecom* and oncogene (*) integrations. The count of other unique integration sites (UIS) is described in the last row (gray bar). (**A**) Integration sites of RSF91-transduced S3F11-Exp bulk population (sample no. 13, IVIM ID 20210311) on day 15 and an expanded clone obtained from low cell density replating. (**B**) Integration sites of RSF91-transduced S3F11+APU-Exp bulk population (sample no. 18, IVIM ID 20220311) on day 15 and day 21 as well as corresponding expanded clones obtained from both low cell density replating time points. (**C**) Normalized enrichment score (NES) after gene set enrichment analysis with SAGA (SAGA-GSEA) of APU-expanded (n = 3) and S3F11+APU-Exp (n = 4) IVIM assays using the bulk population from day 15. Statistical analysis performed with one-way ANOVA and Kruskal-Wallis multiple comparisons test, *p* < 0.05 = *, *p* < 0.01 = *** (**D**) Effect size of NES_RSF91_ versus NES_SIN.LV.EFS_ from SAGA using fresh, S3F11-Exp, or S3F11+APU-Exp cells. Cohen’s d of each condition is shown with 95% confidence interval. The biggest effect between RSF91 and SIN.LV.EFS was observed for S3F11+APU-Exp. (**E**) Cell yield from one mouse after LSK SLAM expansion in S3F11 or S3F11+APU versus after isolating nonexpanded lin- cell for the IVIM assay. Expansion factor of total cell number after 14 days was multiplied by the mean of LSK SLAM cells isolated from one mouse (mean from the last five isolations: 1115 LSK SLAM cells/mouse). Without expansion, one mouse yields cells for only 2–3 IVIM assays (5 × 10^5^–1 × 10^6^ per assay). Expanding LSK SLAM cells in S3F11+APU results in over 400 vials for gene therapy studies.

## Data Availability

Not applicable.

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
