# Peer review of "TGFβ Inhibitor A83-01 Enhances Murine HSPC Expansion for Gene Therapy"

_cells, 2023, doi:10.3390/cells12151978_

Round 1
Reviewer 1 Report
Fleischauer et al showed interesting results supported by an appropriate research design. Authors evaluated cell culture conditions to attenuate differentiation during HSPC culture, by acting on Smad2 pathway through AB3-01 TGF-beta inhibitor. Moreover, authors showed a similar immune phenotype in fresh cells and APU-expanded HSPC transduced with retroviral vector and they test their functionality in two in vitro genotoxicity assays.
The manuscript structure is straightforward, comprehensive and fluent. All the points listed in the abstract have been well-argued in the whole text. Discussion clearly elucidates the reached results. Figures are clear and well-structured.
Reviewer 2 Report
Manuscript by Fleischauer et al:
The authors describe an improved protocol for expansion of murine HSPCs for testing the mutagenicity of viral vectors for gene therapy use.
Throughout the manuscript, qualities and quantities of fresh and expanded cells are compared. In the beginning of the results, the authors mention that expanded (-Exp) cells are always from LSK-SLAM sorted BM. The traditional assays with fresh material is Lin-depleted BM. Is this correct for all figures (except where otherwise specified as in fig. 1F)? It is not always clearly labeled in all figures. Also, I understand the reasoning for this comparison (fresh Lin- vs. -Exp LSK-SLAM), but miss this difference in the discussion part, where several observations are discussed, such as the reduced sensitivity of the IVIM assay with -Exp cells compared to fresh cells. Not just the freshness, but also the source of starting material could have an effect on everything so it is fair to discuss that.
Regarding the last sentence of the results: 'similar behaviour... suitable for genotoxicity assays...' While largely true, there is the reduced IVIM assay sensitivity of -Exp cells compared to fresh, as mentioned in the discussion (line 508). In Suppl. fig. 5 there is a higher replating frequency of fresh cells, also compared to the number of dots representing the negative assays. Can the authors say anything on whether this new expansion protocol is now good enough and can really be used instead of the traditional assay for testing of clinical viral products?
In Fig. 7E (lines 441-445) the authors calculate how many vials of S3F11 and S3F11+APU -Exp cells can be stored from one mouse. How many cells are needed per experiment and what is the assumed fold of expansion for this calculation? I couldn't gather this from the expansion factors in figure 1. One extra sentence on how to get to 429 vials would be good.
Minor points:
line 251 and legend fig. 1B: significant higher expansion of cells in S3F11. What about S3F11T? Also, 4th line of legend 1B: 'S3F11, or S3F11...medium' Is a 'T' missing?
Legend for Figures 1 and 7 are not complete. Legend Fig. 1 D,E,F missing. Fig 7E legend is cut off mid sentence.
Figure 5: colors of the lines: green, orange, blue in figure is inconsistent with those in the legend 5B and in the text lines 334-336.
legend 5B: past tense of split is split (not splitted)
line 385 etc.: replating after 21 days. In the suppl. figures, it is 22 days.
line 387: one immortalized clone; it looks like 2 red triangles in suppl fig. 5. Or did one of those not lead to clones?
Fig. 7E: what is the time of the proposed expansion protocol: one week or two weeks? Good to add this in the illustration.
